# Tidal Effects on Dissolved Organic Matter Dynamics in a Brackish Water Front Adjacent to Yangtze River Estuary

Yasong Wang [†], Niting Peng [†], Zhiliang Liu, Liang Liu, Sishang Pan, Dayu Duan and Yunping Xu *

College of Oceanography & Ecological Science, Shanghai Ocean University, Shanghai 201306, China; yswang@shou.edu.cn (Y.W.); d230200062@st.shou.edu.cn (L.L.)
* Correspondence: ypxu@shou.edu.cn
† These authors contributed equally to this work.

**Abstract:** A brackish water front, where river water meets seawater, is a hotspot for biogeochemical processes. In this study, we examined the quantity and composition of dissolved organic matter (DOM) over a 24 h tidal cycle at a brackish water front near the Yangtze River estuary. Utilizing elemental analysis, fluorescence and ultraviolet spectroscopy, and ultra-high-resolution mass spectrometry, we observed rapid fluctuations in DOM throughout the tidal cycle. The dissolved organic carbon (DOC) and total nitrogen (TN) concentrations ranged from 0.70 to 1.5 mg/L and 0.43 to 0.94 mg/L, respectively. Water samples during low tide exhibited a higher fractional abundance of CHON (17.2 ± 0.1% vs. 14.6 ± 0.4%), CHOS (14.6 ± 4.5% vs. 9.1 ± 3.1%), and CHONS (1.6 ± 0.5% vs. 0.5 ± 0.3%) formulas, and a higher aromatization and average molecular weight, which is consistent with a stronger terrestrial influence. In contrast, at high tide, the water samples contained a greater abundance of CHO compounds (75.7 ± 3.8% vs. 66.5 ± 4.1%), a humic-like fluorescent C1 component, and carboxyl-rich alicyclic molecules (CRAMs), indicating a greater release of refractory DOM from resuspended sediments. However, variations in the DOC concentrations and several optical spectral parameters did not correlate with the changes in the salinity and tidal height. The results of the principal component analysis revealed different controls on specific fractions of DOM that are related to variable DOM sources or biogeochemical processes. The complexity of DOM dynamics underscores the necessity of elucidating DOM compositions at varying levels to enhance our understanding of carbon cycling in estuarine and coastal ecosystems.

**Keywords:** dissolved organic matter; tidal cycle; brackish water front; FT-ICR MS; Yangtze River

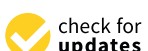

## 1. Introduction

Global rivers transport approximately 0.30 ± 0.14 Pg of dissolved organic carbon (DOC) and 0.18 ± 0.04 Pg of particulate organic carbon into the oceans each year, playing a crucial role in the global carbon cycle [1,2]. In estuarine environments, tidal hydrodynamics exert a profound influence on biogeochemical processes and ecosystems [3–5]. Variations in tidal flow velocity and tidal height can change the migration behaviors of sediments, leading to varying degrees of resuspension and movement of sediments, which in turn alters the release and uptake of nutrients and organic carbon in estuarine areas [6,7]. Recent studies have identified sediment resuspension as a significant source of refractory dissolved organic matter (DOM) in overlying water bodies [8]. Tidal simulation experiments indicate that changes in tidal intensity can modify the release of sedimentary organic matter into the water column and influence the structure of microbial communities in the water,

thereby affecting the fluorescence intensity of DOM [9]. Takasu et al. [10] compared DOM and particulate organic matter in three estuaries on Kyushu Island, Japan, revealing that particulate organic matter is significantly affected by tidal fluctuations, while DOM is primarily controlled by the mixing of fresh water and saltwater. Similar findings were also found in the Hudson River estuary [11]. Although the scientific community has widely acknowledged the critical role of tides in the biogeochemical cycling of estuarine regions, there remains a limited understanding of the rapid variations in organic matter over tidal cycles [12].

The Yangtze River (or Changjiang River) is the third longest river in the world and the largest river in China, discharging substantial amounts of freshwater, suspended sediments, and nutrients into the East China Sea [13,14]. Song et al. [15] investigated the variations in DOM and nutrients during spring, moderate, and neap tides in the Yangtze River estuary, revealing significant changes in the concentrations of DOM and monosaccharides in relation to tidal fluctuations. These variations may be attributed to the physical mixing of saline and freshwater masses, the adsorption/desorption of particulate matter, and microbial degradation processes. Please change to: The extreme flood events can lead to a substantial input of chromophoric dissolved organic matter (CDOM) into the Yangtze River estuary and adjacent sea [16], while the terrestrial organic matter also undergoes changes in molecular composition and structure as it passes through the turbidity zone [17]. These studies suggest that changes in hydrodynamic conditions can cause alterations in DOM molecules on both spatial and temporal scales in the Yangtze River estuary.

Despite a relatively comprehensive understanding of the types, structures, and seasonal variations in DOM in the Yangtze River estuary, previous studies have primarily focused on larger spatial scales (tens to hundreds of kilometers) and longer temporal scales (such as seasonal and interannual variations) [18,19], with limited research on rapid changes at smaller scales. This study aims to investigate the brackish water front at a site near the Yangtze River estuary, tracking the changes in the DOM concentration and composition during tidal cycle to shed light on the tidal effect on carbon cycling in the river-dominated margin.

## 2. Study Area and Sampling

### 2.1. Study Area

The sampling site is located in the northeastern coastal waters of Zhejiang Province, specifically in the central region of the Zhoushan Archipelago, south of the Yangtze River estuary (Figure 1). The water depth ranges from 10 to 20 m. This area is characterized by a subtropical monsoon climate, with warm and humid conditions throughout the year. The major currents are the Changjiang Diluted Water (CDW), Zhe-Min Coastal Current (ZMCC), and Kuroshio Current (KC). The tidal regime in the waters near Zhoushan is classified as irregular semi-diurnal, exhibiting significant variations in the tidal range and unequal diurnal tides. The sediment type is mud. The Yangtze River is the most important source of sediments, with the annual average sediment flux of around 100 Mt $yr^{-1}$ and an organic carbon flux of around 4.8 Mt $yr^{-1}$ [20]. After the Yangtze River, the Qiantang River is the second largest river in the study area, discharging 6.7 Mt of sediments into the sea.

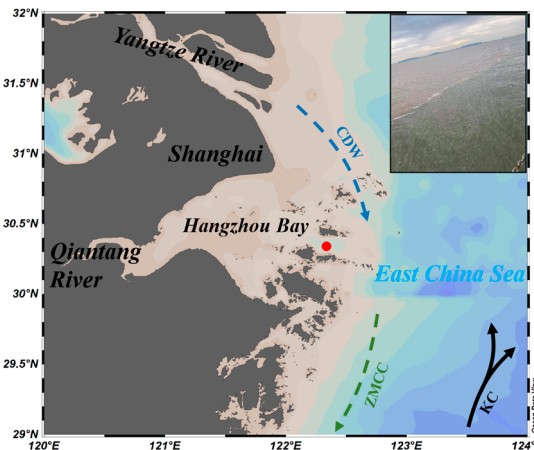

**Figure 1.** Map showing the study area and sampling site. CDW: Changjiang Diluted Water; ZMCC: Zhe-Min Coastal Current; KC: Kuroshio Current. The panel (top right) presents a photograph taken onsite, clearly illustrating the interface between river water and seawater.

### 2.2. Sampling and Pretreatment

From May 11 to 12, 2023, we collected surface seawater samples (1 m in depth) in the Zhoushan Archipelago of the East China Sea (30°20′30″ N, 122°20′20″ E) aboard the RV Songhang (Shanghai Ocean University). Sampling was conducted at 1 h intervals. Following collection, all seawater samples were preserved in low-temperature (~4 °C), light-protected conditions and transported back to the laboratory within 48 h. Concurrently, in situ measurements of surface water physicochemical parameters, including temperature (17.2–19.6 °C), pH (7.97–8.01), and salinity (25.5–28.7 ppt), were conducted using portable instruments.

Upon returning to the laboratory, seawater samples were immediately filtered using 0.7 µm glass fiber filters (GF/F, Thermo Scientific, Waltham, MA, USA) that were pre-heated at 450 °C for 4 h. A portion of the filtration was transferred to 100 mL HDPE bottles that were pre-washed with ultrapure water, hydrochloric acid, and, subsequently, ultrapure water. All samples were stored in refrigeration for analyses of the DOC, CDOM, and fluorescent dissolved organic matter (FDOM). For further analysis using Fourier transform ion cyclotron resonance mass spectrometry (FT-ICR MS), the filtrate was first subjected to solid-phase extraction to isolate 10 L of acidified DOM samples (Agilent Bond Elut PPL 200 mg, Agilent Co., Santa Clara, CA, USA), followed by elution with methanol [21].

### 2.3. Analytical Methods

#### 2.3.1. Total Carbon and Nitrogen Analyses

A 15 mL sample of the filtrate was acidified to pH 2, after which the concentrations of DOC and total nitrogen (TN) were analyzed using a Shimadzu TOC-L analyzer. The method can be found in detail in Wang et al. [22].

#### 2.3.2. Ultraviolet-Visible (UV-Vis) Spectroscopy Analysis

Measurements were conducted using a Shimadzu UV-2600 dual-beam UV-vis spectrophotometer (Shimadzu, Japan). A quartz cuvette with a 10 cm optical path length was employed, with Milli-Q purified water (18.2 MΩ.cm; Merck KGaA, Darmstadt, Germany) as the reference. The scanning wavelength range was set from 200 to 800 nm, with a scanning interval of 1 nm. Ultra-pure water blanks were measured every four samples to ensure instrument stability. The following three parameters were calculated based on the UV-vis spectral data: $a_{325}$, SUVA254, and S275-295. The parameter $a_{325}$ represents the absorbance coefficient at 325 nm, indicating the relative concentration of CDOM [23]. SUVA254 is

defined as the absorbance at 254 nm divided by the DOC concentration, reflecting the aromaticity of the DOM [24]. The S275-295 parameter indicates the slope of the spectral absorbance from the 275–295 nm range to the 350–400 nm range, which is indicative of the average molecular weight of DOM; a larger S275-295 value suggests a smaller molecular weight [25].

### 2.3.3. Fluorescence Spectroscopy Analysis

The three-dimensional fluorescence spectra of the samples were measured using a Hitachi F-7000 fluorescence spectrophotometer. The instrument parameters included a 1 cm optical path quartz cuvette, an excitation wavelength range of 240–450 nm, an emission wavelength range of 250–550 nm, and a scanning step size of 5 nm. Milli-Q ultra-pure water was used as a blank for scattering correction, and the Raman units (R.U., $nm^{-1}$) were utilized for calibration. To minimize instrument error, ultra-pure water blanks were measured every six samples to ensure stability. Key fluorescence parameters included the Humification Index (HIX), Biological Index (BIX), and Fluorescence Index (FI) [26,27]. An HIX value less than 3 indicates a weak humification degree of DOM, suggesting a recent microbial or bacterial origin; an HIX value between 3 and 6 indicates an intermediate humification degree, with reduced recent autochthonous characteristics; and an HIX value greater than 6 suggests significant contributions from terrestrial inputs, with strong humification [28]. The BIX value reflects the intensity of the aquatic biological activity. A BIX value less than 0.8 indicates that the DOM is primarily of terrestrial origin with minimal autochthonous contributions, while a BIX value greater than 1 suggests that the DOM is predominantly derived from algae or bacteria, indicating significant autochthonous characteristics [29]. The FI typically increases with the proportion of aquatic DOM; an FI value greater than 1.9 usually indicates that the DOM primarily originates from microbial activity within the water body, whereas an FI value less than 1.4 suggests a predominant contribution from external sources (e.g., terrestrial inputs) [29,30].

Additionally, parallel factor analysis (PARAFAC) was conducted to identify the fluorescent components of the DOM samples [31]. A modeling analysis was performed on 30 samples (excluding outliers), with the model constrained to non-negative values. Repeated computations were conducted for models comprising three to seven components, resulting in a three-component model. This model was compared with the OpenFluor database (https://openfluor.org; accessed on 23 September 2023) based on the Ex/Em wavelengths, achieving a fitting degree of TCCex, em > 0.95 [32].

### 2.3.4. FT-ICR MS Analyses

The molecular composition characteristics of the DOM were determined using electrospray ionization (ESI) coupled with FT-ICR MS. This specific procedure was conducted at the State Key Laboratory of Heavy Oil at the China University of Petroleum (Beijing), utilizing a 9.4 T Apex-Ultra FT-ICR MS interfaced with a German Apollo II ESI source, analyzing the FT-ICR MS samples in the negative ion mode. The instrument parameters were set as follows: signal-to-noise ratio (S/N) $\geq 6$ and mass error for molecular formulas < 0.5 ppm, with a sample flow rate of 250 $\mu L\ h^{-1}$. The molecular characteristic parameters were established in van Krevelen distribution plots (elemental ratios: H/C vs. O/C) to assess the overall composition. The modified aromaticity index (AImod), double bond equivalent (DBE), average molecular weight (MW), and molecular labile compounds (MLB, H/C $\geq$ 1.5) were calculated according to previous reports [33,34]. The molecular formulas were further classified into different compounds as follows: polycyclic aromatics (PCAs, AImod $\geq$ 0.67), polyphenols (PPs, 0.5 $\leq$ AImod < 0.67), highly unsaturated compounds (HUCs, AImod < 0.5 and H/C < 1.5), unsaturated aliphatic compounds (UACs,

$1.5 \leq H/C < 2.0$ and $N = 0$), and peptides (PTs, $1.5 \leq H/C < 2.0$ and $N > 0$), and carboxyl-rich alicyclic molecules (CRAMs, $0.30 < DBE/C < 0.68$; $0.20 < DBE/H < 0.95$; and $0.77 < DBE/O < 1.75$) [35,36].

### 2.4. Data Processing

The FDOM components were analyzed using MATLAB 2018a for spectral analysis, while other graphical representations were generated using Origin 2021 and ODV (Ocean Data View) software [37]. Tidal data were obtained from the maritime service website (https://www.cnss.com.cn/tide; accessed on 21 August 2023). We used tidal data from the Daishan Hydrological Station, which is close to our sampling site.

## 3. Results

### 3.1. Tidal Variation and Salinity

The variations in the tidal levels clearly illustrate the characteristics of the irregular semi-diurnal tide in the Zhoushan coastal area (Figure 2a). The first high tide occurred at 7–8 h (with the initial sampling time as the reference point), reaching a height of 3.3 m, while the second high tide was observed at 20 h, with a height of 2.2 m. Notably, the second high tide was significantly lower than the first one. This irregular semi-diurnal tide is influenced by factors such as the underwater topography of the estuaries of the Yangtze River and Qiantang River, as well as river discharge. Correspondingly, the salinity of the surface water exhibited significant variations during the sampling period, ranging from 25.5 to 28.7. The trend in salinity changes was generally consistent with that of tidal heights, showing an increase in salinity during rising tides and a decrease during falling tides. The maximum salinity was recorded at the first high tide, whereas the salinity during the second high tide was notably lower than that at the first peak (see Figure 2a). This trend effectively reflects the relative contributions of seawater and river water, indicating that during rising tides, seawater intrusion intensifies, leading to increased salinity, while during falling tides, seawater intrusion diminishes, resulting in decreased salinity.

### 3.2. Variations in DOM Concentrations and UV-Vis Optical Characteristics

The DOC concentration varied between 0.70 and 1.5 mg/L, with an average of $0.96 \pm 0.21$ mg/L. The highest DOC concentration was observed at low tide, while the lowest concentration occurred at high tide (Figure 2b). The TN concentration ranged from 0.43 to 0.94 mg/L, with an average of $0.71 \pm 0.10$ mg/L. The trend in TN concentration exhibited an inverse relationship with tidal levels, although there was a slight temporal lag. During rising tide, TN concentrations gradually decreased, reaching their minimum at peak tide, followed by an increase during falling tide (Figure 2c).

The $a_{325}$ varied from 0.88 to 3.65 $m^{-1}$, showing rapid changes around the first high tide, likely due to strong hydrodynamic forces that increased disturbances between the water column and surface sediments, resulting in a significant increase in the $a_{325}$ values (Figure 2d). However, the $a_{325}$ remained relatively constant before and after the subsequent high tide. The BIX ranged from 0.89 to 1.04, with an average of $0.92 \pm 0.03$ (Figure 2e). Overall, the BIX changes did not show a clear correlation with tidal fluctuations, with the maximum value occurring during a rapidly rising tide. The HIX ranged from 0.41 to 0.57, with an average of $0.52 \pm 0.04$ (Figure 2f). The HIX values remained relatively stable, although two distinct low values were observed during the increase to maximum tide and during the subsequent decline.

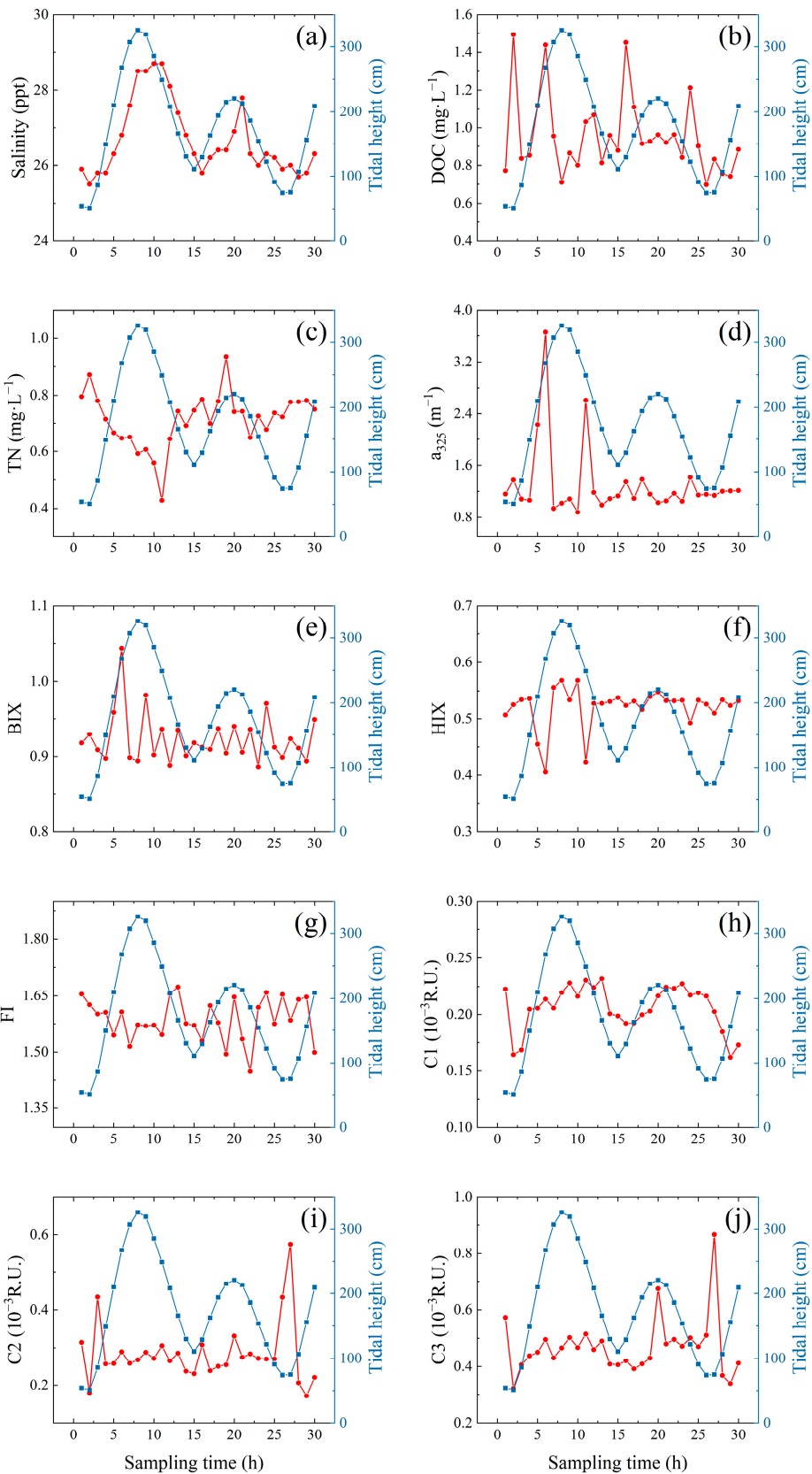

**Figure 2.** Variations in the tidal height, salinity, and DOC and TN contents, as well as optical parameters, in water samples during one tidal cycle: (**a**) salinity; (**b**) DOC concentration; (**c**) TN concentration; (**d**) $a_{325}$; (**e**) BIX; (**f**) HIX; (**g**) FI; (**h**) C1; (**i**) C2; (**j**) C3.

### 3.3. Variations in Fluorescence Optical Characteristics

The FI varied from 1.49 to 1.67, with an average of 1.59 ± 0.06 (Figure 2g). The FI values were generally stable, without significant trends observed. Based on the analysis of 30 samples (excluding outliers), a PARAFAC modeling identified the following three fluorescent components (Figure 3): C1 (Ex/Em = 245/446 nm), which is associated with humic components mainly derived from terrestrial organic matter [27], and C2 (Ex/Em = 275/325 nm) and C3 (Ex/Em = 240/354 nm), which are related to protein-like components primarily associated with aquatic biological activity (such as microbial degradation or biological residues); however, in areas with intense human activity and severe pollution, terrestrial inputs can also become a significant source of protein-like components [27,29,38]. The C1 component varied from 0.16 to 0.23 × $10^{-3}$ R.U., with an average of 0.21 ± 0.02 × $10^{-3}$ R.U., showing a general trend consistent with tidal fluctuations (i.e., a high level at high tide and a low level at low tide), albeit with a slight temporal lag (Figure 2h). The C2 component ranged from 0.17 to 0.57 × $10^{-3}$ R.U., with an average of 0.28 ± 0.08 × $10^{-3}$ R.U., while the C3 component ranged from 0.32 to 0.87 × $10^{-3}$ R.U., with an average of 0.47 ± 0.10 × $10^{-3}$ R.U. (Figure 2i,j). The trends for the C2 and C3 components were generally similar, remaining low before and after the maximum tide, with occasional high values appearing around the subsequent high tide.

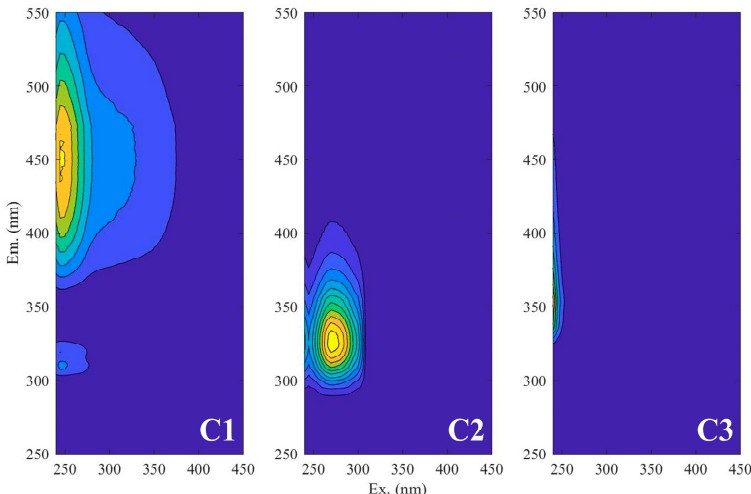

**Figure 3.** Three fluorescence components (C1, C2, and C3) were identified based on EEMs-PARAFAC: humic-like component C1 (Ex/Em = 245/446 nm); protein-like component C2 (Ex/Em = 275/325 nm); protein-like component C3 (Ex/Em = 240/354 nm).

### 3.4. Variations in Molecular Formulas Revealed by FT-ICR MS

Based on the optical spectral data, we selected six samples (#1, #2, #13, #16, #23, and #29) for the FT-ICR MS analysis. Among these, samples #2, #16, and #29 were collected near low tide, while samples #1, #13, and #23 were collected near high tide. Figure 4 shows typical Van Krevelen diagrams for the high-tide and low-tide samples based on the H/C and O/C ratios of the DOM formulas. The molecular compositions detected through FT-ICR MS are summarized in Table 1. At high tide, an average of 3766 ± 473 formulas were detected, comprising 1675 ± 35 CHO formulas, 1344 ± 154 CHON formulas, 660 ± 232 CHOS formulas, and 87 ± 71 CHONS formulas. While at low tide, an average of 6132 ± 445 formulas was detected, including 2109 ± 219 CHO formulas, 2389 ± 278 CHON formulas, 1219 ± 291 CHOS formulas, and 414 ± 123 CHONS formulas. Overall, the numbers of total formulas and each type of formula were consistently greater in the low-tide samples compared to the high-tide samples.

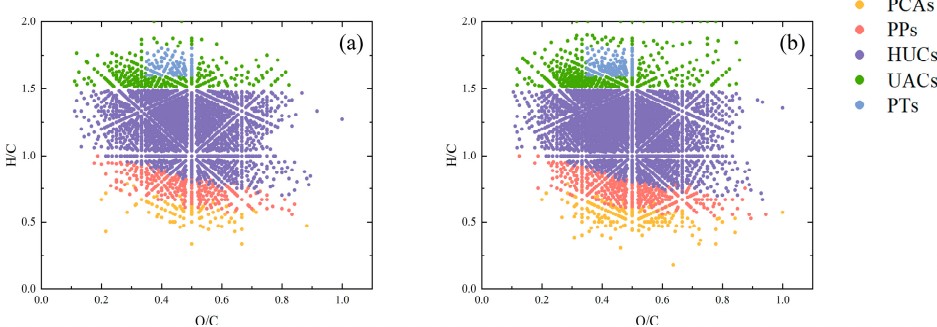

**Figure 4.** van Krevelen diagrams: (**a**) high-tide DOM; (**b**) low-tide DOM.

**Table 1.** Numbers of the total and different types of formulas identified by FT-ICR MS. Bold denotes the high-tides samples.

| Formulas | #1 | #2 | #13 | #16 | #23 | #29 |
|---|---|---|---|---|---|---|
| Total | 3606 | 6602 | 3394 | 6076 | 4300 | 5717 |
| CHO | 1636 | 2084 | 1684 | 2339 | 1705 | 1904 |
| CHON | 1304 | 2552 | 1214 | 2548 | 1514 | 2068 |
| CHOS | 612 | 1419 | 457 | 885 | 913 | 1354 |
| CHONS | 54 | 547 | 39 | 304 | 168 | 391 |

Regarding the fractional abundance of different molecules, the high-tide samples exhibited a higher fractional abundance of CHO compared to the low-tide samples ($75.7 \pm 3.8\%$ vs. $66.5 \pm 4.1\%$); however, the high-tide DOM had a lower fractional abundance of CHON ($14.6 \pm 0.4\%$ vs. $17.2 \pm 0.1\%$), CHOS ($9.1 \pm 3.1\%$ vs. $14.6 \pm 4.5\%$), and CHONS ($0.5 \pm 0.3\%$ vs. $1.6 \pm 0.5\%$) than the low-tide samples (Figure 5a–d). The statistical analyses suggest significant difference in the fractional abundance of CHON between the high- and low-tide DOM but no significant difference in fractional abundance of the CHO, CHOS, and CHONS.

The averaged MW, DBE, and AImod of the DOM at high tide were $437.3 \pm 3.9$, $8.56 \pm 0.04$, and $0.22 \pm 0.0$, respectively. In contrast, the MW, DBE, and AImod of the DOM at low tide were $457.2 \pm 3.0$, $8.92 \pm 0.10$, and $0.22 \pm 0.0$, respectively (Figure 5e–g). Overall, the low-tide DOM exhibited significantly higher values of MW and DBE compared with the high-tide DOM, whereas the AImod values were slightly lower in the low-tide DOM than that in the high-tide DOM.

The Van Krevelen diagrams further classify the DOM formulas into the following six categories: polycyclic aromatics (PCAs), polyphenols (PPs), highly unsaturated compounds (HUCs), unsaturated aliphatic compounds (UACs), and peptides (PTs) [35,36]. We combined the PCAs and PPs into a single group and the UACs and PTs into another group because the former are aromatic compounds, whereas the latter are non-aromatic compounds [17]. The percentages of PCAs+PPs, HUCs, and UACs+PTs during ng high tide were $2.17 \pm 0.17\%$, $86.4 \pm 0.4\%$, and $8.3 \pm 0.1\%$, respectively, while during low tide, these percentages were $2.62 \pm 0.09\%$, $86.0 \pm 0.5\%$, and $8.4 \pm 0.3\%$, respectively (Figure 5h–j). Correspondingly, the percentages of CRAMs were $66.6 \pm 1.5\%$ in the high-tide DOM and $63.8 \pm 1.5\%$ in the low-tide DOM (Figure 5k). The high-tide DOM exhibited a significantly lower level of PCAs+PPs% compared with the low-tide DOM but slightly higher levels of HUGs% and CRAMs%.

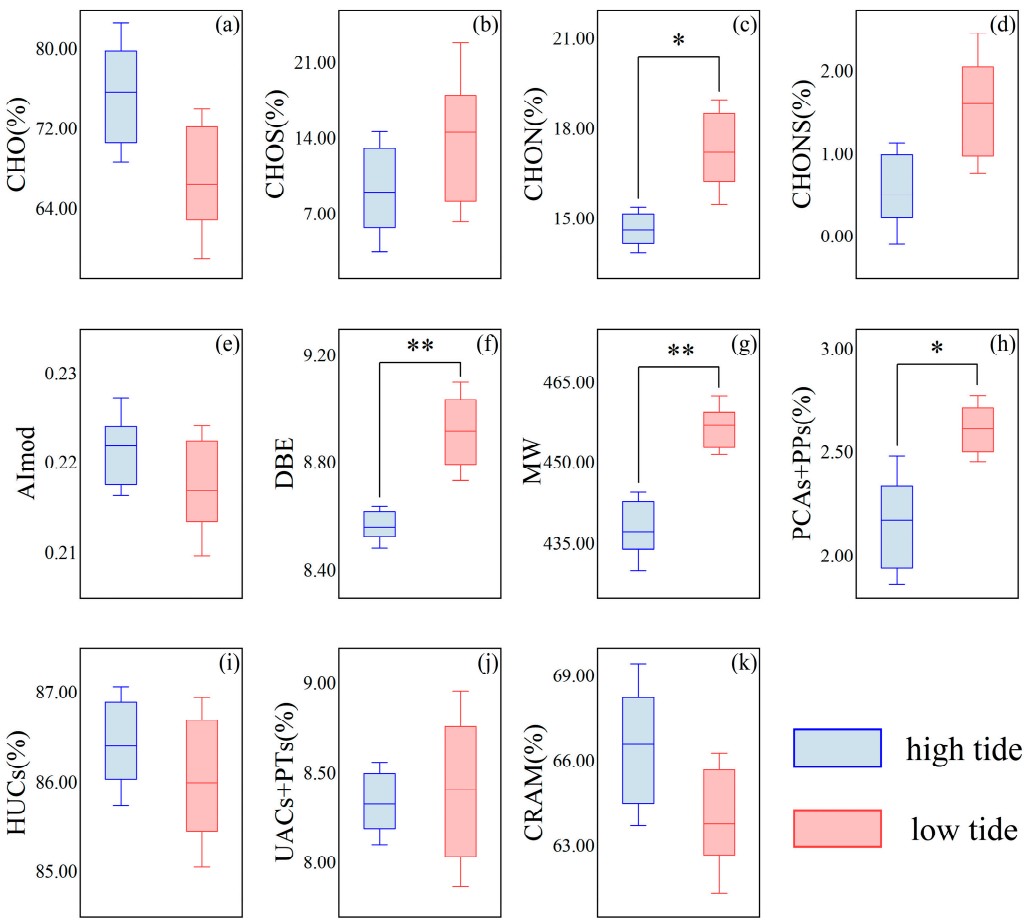

**Figure 5.** Comparison of multiple indicators between the high- and low-tide DOM based on the FT-ICR MS data: (**a**) CHO%; (**b**) CHON%; (**c**) CHOS%; (**d**) CHONS%; (**e**) AImod; (**f**) DBE; (**g**) averaged MW; (**h**) PCAs+PPs%; (**i**) HUGs%; (**j**) UACs+PTs%; (**k**) CRAM%. *, ** Denote significant differences at the levels of $p < 0.05$ and $p < 0.01$, respectively. All indicators were calculated based on the intensity of each formula.

## 4. Discussion

### 4.1. Different Behaviors of DOC and TN Concentrations Within the Tidal Cycle

The TN concentration reached its highest level during low tide and the lowest levels during high tide (Figure 2c). It had significant negative correlations with seawater salinity and tidal height ($p < 0.01$; Figure 6), indicating that TN exhibited a conservative behavior during physical mixing between river water and seawater [39]. The higher TN concentration during low tides and lower TN concentration during high tides suggest that the TN in the study area is primarily derived from terrestrial inputs. During low tide, enhanced river influence contributes to more nitrogen, whereas during high tide, increased seawater intrusion leads to the dilution of the TN concentration. This observation is consistent with previous reports on nutrient distributions in Hangzhou Bay and its adjacent sea. It is reported that TN concentrations in the Yangtze River estuary range from 1.78 to 2.46 mg/L [40,41], while concentrations in the Qiantang River estuary can be as high as 2.50 to 2.90 mg/L, significantly exceeding the TN concentration observed on the East China Sea continental shelf (0.004~0.2 mg/L) [42].

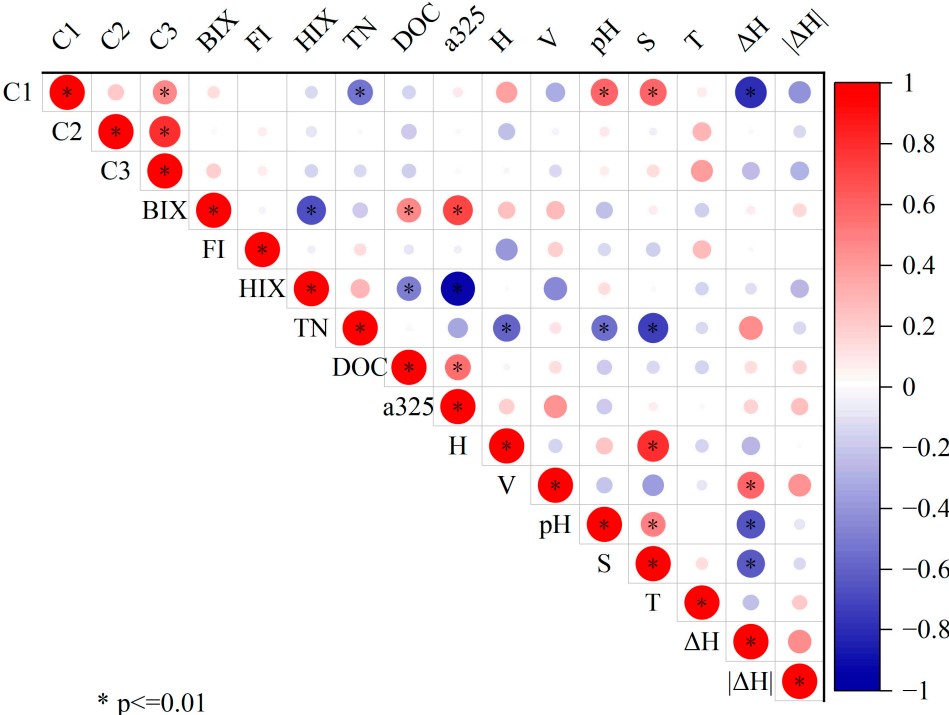

**Figure 6.** Heatmap of the correlations between environmental factors and optical parameters in surface water samples during tidal cycling.

Compared with the TN, the relationship between the DOC concentration and tidal variation is more complex. There was no significant correlation between the DOC concentration and seawater salinity or tidal height ($p > 0.05$; Figure 6). This suggests that the DOC concentrations are not solely controlled by riverine input but are also influenced by in situ biogeochemical processes. Strong tidal forces may lead to the resuspension of local sediments, which can release DOC into the water or adsorb DOC from water bodies. Meanwhile, the changes in water turbidity associated with tides could affect primary productivity via controlling nutrients and light availability, which then influence DOC concentrations. The relationship between turbidity and phytoplankton productivity, which is reflected by chlorophyll-a, was observed in tidal-influenced Hangzhou Bay [43]. Additionally, DOC comprises various compounds with differing reactivities, which may undergo selective degradation within the tidal cycles. These factors contribute to the significant yet complex variations in DOC throughout the tidal period.

### 4.2. Variable DOM Compositions Within Tidal Cycle

The spectral parameter $a_{325}$ exhibits a positive correlation with the DOC concentration ($R^2 = 0.29$, $p < 0.01$; Figure 6), indicating that CDOM is a significant component of bulk DOM. However, their correlation is not very strong, suggesting that DOC concentrations are also affected by other factors. The variable mixing of terrestrial and marine DOM is reflected in several spectral indicators. Notably, the terrestrial humic component C1 consistently constituted an important part of the FDOM, averaging $21.92 \pm 2.59\%$, while the protein-like components C2 and C3, primarily derived from aquatic organisms, averaged $29.28 \pm 3.51\%$ and $48.80 \pm 2.54\%$, respectively. Additionally, during tidal periods, the FI ($1.59 \pm 0.06$; Figure 2g) falls between terrestrial DOM (FI < 1.4) and marine DOM (FI > 1.9) [29,30], while the BIX values ($0.92 \pm 0.03$) were also situated between DOM that is primarily influenced by terrestrial inputs (<0.8) and high-productivity aquatic DOM (>1.0) [29]. Collectively, these observations suggest that during tidal processes, the brackish water front consistently exhibits a mixed contribution from terrestrial and marine DOM sources.

The molecular information obtained from the FT-ICR MS further confirms the variations in DOM during the tidal cycling, with sulfur- and nitrogen-containing compounds (CHOS, CHON, and CHONS) in the DOM being higher at low tide compared to high tide (Figure 5b–d). Previous studies found that the abundance of CHOS and CHONS compounds decreases along the pathway from freshwater to offshore regions of the Yangtze River [44]. Thus, the results from the FT-ICR MS reflect the impact of freshwater and seawater mixing on the molecular compositions of DOM. At low tide, the increased riverine influence leads to a relative increase in sulfur- and nitrogen-containing compounds, while at high tide, the strengthening influence of seawater results in a relative decrease in these compounds. Consequently, the relative abundance of sulfur- and nitrogen-containing compounds can serve as a molecular indicator of terrestrial inputs.

Additionally, the low-tide DOM contained significant higher levels of averaged DBE, MW, and PCAs+PPs% compared with the high-tide DOM (Figure 5f–h). This further indicates that the more pronounced riverine input during low tide brings in greater amounts of anthropogenic-derived dissolved black carbon (DBC) and other terrestrial aromatic compounds (e.g., lignin), which is consistent with the higher relative abundance of CHON, CHONS, and CHOS compounds observed at the low tide. The higher DBE and MW of the low-tide DOM also supports that the terrestrial DOM is characterized by more high-molecular-weight and unsaturated compounds. However, it is noteworthy that the AImod index, also indicative of aromatic compounds, displays slightly higher levels, despite no significant difference, in the high-tide DOM compared with the low- tide DOM (Figure 5e). This trend is opposite to that of the PCAs+PPs% (Figure 5h). This inconsistency between AImod and PCAs+PPs% may be explained by the following two factors: (1) the AImod index reflects the average aromaticity of all detectable DOM formulas by FT-ICR MS from monoaromatic to condensed aromatic rings, while PCAs and PPs are representative of compounds containing polycyclic aromatic hydrocarbons, which differ in their indicative properties; and (2) the higher in situ primary productivity during low tide results in a greater contribution from in situ phytoplankton-derived DOM that is depleted in aromatic carbon, diluting the average aromaticity of the DOM, a hypothesis supported by the higher DOC concentration data observed at low tide (Figure 2b). Compared to UV and fluorescence spectral indices (e.g., FI, HIX, and BIX; Figure 2), molecular indices derived from FT-ICR MS (such as CHOS%, CHONS%, PCAs%, and PPs%) can more sensitively reflect changes in the DOM's composition during tidal processes. Therefore, we recommend that studies of DOM in complex environments should not rely solely on simpler spectral parameters.

### 4.3. Different Controls on DOM Components Within a Tidal Cycle

Despite the widespread use of ultraviolet and fluorescence spectral parameters to indicate the sources and transformations of DOM in the environment, they did not exhibit significant trends during the tidal cycle (Figure 2). Typically, the HIX and C1 indices, which usually indicate the intensity of the terrestrial input, did not show maximum values during low tide (associated with the strongest freshwater influence); instead, a lower value for C1 was recorded. Similarly, indices such as BIX, FI, C2, and C3, which indicate contributions from aquatic organisms, did not reach maximum values during high tide (corresponding to the strongest seawater intrusion), with the protein-like component C2 showing particularly low values at high tide. Statistical analyses reveal that, aside from a significant correlation with C1, the salinity did not show any significant correlations with C2, C3, BIX, FI, or HIX (Figure 7). This further demonstrates that the composition of the DOM is influenced not only by the physical mixing of terrestrial and marine sources but also by biogeochemical processes. Notably, $a_{325}$ exhibited two peaks during rapid tidal changes near maximum tide levels. A weak positive correlation was observed between $a_{325}$ and tidal current velocity

($R^2 = 0.19$, $p < 0.05$, n = 30), with greater deviations occurring during periods of high current velocity before and after spring tides. We suggested that the rapid changes in tide could cause significant resuspension of sediments, leading to the release of particulate organic matter into the water column as CDOM. Previous studies have also reported substantial differences in the DOM's molecular composition during drainage periods (i.e., when the water flow velocity is higher) [45]. However, the HIX values showed two minimums during periods of rapid tidal changes (Figure 2f). A weak negative correlation was found between the HIX and tidal current velocity ($R^2 = 0.20$, $p < 0.05$, n = 30), with greater deviations occurring during periods of high current velocity around spring tides. The inconsistency between the HIX and $a_{325}$ suggests that sediment resuspension may have released CDOM with lower humification, likely derived from less humified organic matter from aquatic organisms. The concurrent high values of BIX further support this hypothesis (Figure 2e). Future studies should focus on simulating sediment resuspension experiments to elucidate the processes of DOM release and absorption between sediments and overlying water.

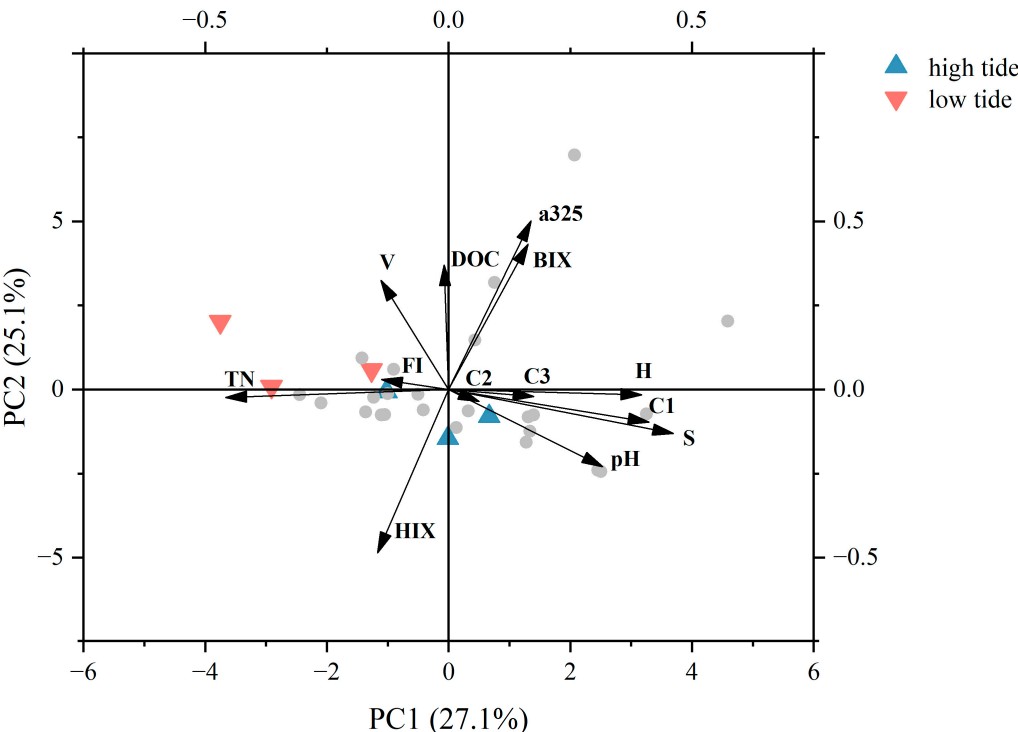

**Figure 7.** PCA results for the tidal samples based on the first two principal components (PCs) of the environmental factors and DOM parameters. Triangles denote the high-tide and low-tide DOM samples selected for the FT-ICR MS analyses.

The proportion of the humic-like C1 component showed a weak positive correlation with water salinity ($R^2 = 0.33$, $p < 0.01$, n = 30). Because the humic-like C1 component is usually thought to derive from terrestrial DOM, its positive relationship with salinity indicates that changes in the C1 component at the brackish water front are not solely controlled by terrestrial inputs. However, the pattern of the variation in C1 closely mirrored that of tidal changes, as follows: decreasing at low tide and increasing at high tide, with a slight temporal lag. In order to further assess the hydrodynamic effect on DOM, we defined the indicator $\Delta H$ as the rate of change in the tidal height between two consecutive sampling times ($\Delta H = H_{n+1} - H_n$). C1 exhibited a significant negative correlation with $\Delta H$ ($R^2 = 0.64$, $p < 0.01$, n = 30) and a weak negative correlation with the absolute value of $\Delta H$ ($R^2 = 0.17$, $p < 0.05$, n = 30). This further supports the idea that tidal disturbances and water flow erosion lead to the release of terrestrial humic components from sediments, resulting

in increased values of these components during strong tides. The rapid movement of large volumes of water leads to significant transport, thereby creating patterns of variation that closely resemble those of tidal changes.

Principal component analysis (PCA) of the environmental parameters, along with the DOM, CDOM, and FDOM, revealed that the first two principal components explained 52.2% of the variance (Figure 7), with PC1 contributing 27.1% and PC2 contributing 25.1%. PC1 exhibited positive correlations with environmental variables such as water pH, salinity (S), and tidal height (H), as well as with humic-like component C1 and protein-like components C2 and C3. Conversely, PC1 showed negative correlations with the TN concentration and FI. Given that salinity and tidal height are sensitive indicators of freshwater–seawater mixing and that nitrogen in the study area primarily originates from riverine inputs [39], it can be inferred that PC1 is predominantly influenced by the change in freshwater and saltwater mixing associated with the tidal cycle. The positive axis of PC1 reflects the strength of the seawater influence, while the negative axis indicates the strength of the freshwater influence.

In contrast, PC2 was positively correlated with the DOC concentration, $a_{325}$, BIX, and tidal current velocity (V), but it had a negative correlation with HIX (Figure 7). Since the DOC concentration is partly related to the primary productivity and BIX is indicative of biological activity in the water, we hypothesize that PC2 is mainly controlled by the marine biological activity in the study area. Additionally, the observed negative correlation between HIX and PC2 suggests that the HIX is not predominantly influenced by terrestrial inputs (as reflected by PC1) but is instead regulated by in situ processes, such as sediment resuspension or the transformation of labile DOM by marine microorganisms.

Furthermore, the PCA plot (Figure 7) reveals a distinct separation between the low-tide and high-tide DOM. The high-tide DOM exhibits higher loadings on PC1 but lower loadings on PC2, whereas the low-tide DOM shows the opposite pattern, with higher loadings on PC2 and lower loadings on PC1. These differences imply that the composition of the DOM varies throughout the tidal cycle, with the high-tide DOM being influenced more by seawater intrusion and microbial transformation, while the low-tide DOM is more affected by freshwater inputs and recent biological productivity. Overall, the PCA results, based on multiple optical and environmental parameters, clearly demonstrate that the DOM in the water is simultaneously influenced by changes in marine-terrestrial mixing, variations in marine productivity, and the resuspension of suspended particulate matter. This underscores that organic carbon stored in shallow coastal sediments may not always serve as a long-term carbon sink due to significant resuspension. In addition to particulate organic carbon, sediment pore water at the land–ocean interface constitutes a substantial pool of DOC with high reactivity [46]. The release of porewater into the overlying water can influence the concentration, reactivity, and cycling of DOC within water columns, particularly in large river estuaries that are strongly affected by tidal forces, coastal currents and anthropogenic activities. Future research will focus on investigating the mechanisms and extent of organic carbon exchange between seafloor sediments and overlying water, considering both natural factors (e.g., tidal cycles and coastal currents) and anthropogenic influences (e.g., shipping, fishery, and trawling).

## 5. Conclusions

This study investigated variations in the DOM in the waters surrounding the Zhoushan Archipelago, influenced by freshwater discharge from the Yangtze River, over tidal cycles. Utilizing elemental analysis, ultraviolet and fluorescence spectroscopy, and high-resolution mass spectrometry (FT-ICR MS), the results indicate significant differences in the chemical properties of the DOM correlated with tidal changes.

(1) The DOC concentration did not exhibit a tidal periodicity, while the TN concentration demonstrated a negative correlation with tidal height and salinity. This suggests that nitrogen levels in the water primarily originate from riverine inputs and exhibit conservative behavior during river–sea mixing, whereas the DOC concentration is influenced by biogeochemical processes beyond physical mixing.

(2) The DOM at low tide displayed higher percentages of nitrogen- and sulfur-containing compounds (CHON%, CHOS%, and CHONS%), increased aromatization (as indicated by PACs+PP%), and higher molecular weight. These findings align with those regarding the Yangtze River–East China Sea continuum, indicating a stronger influence of riverine inputs during low tide. Conversely, high-tide samples contained a greater abundance of CHO%, a humic-like fluorescent C1 component, and CRAM, suggesting a more pronounced influence of marine-derived organic matter or an increased release of refractory DOM from resuspended sediments.

(3) Variations in the optical spectral parameters (such as BIX, HIX, $a_{325}$, and protein-like fluorescent C2 and C3 components) did not correlate with tidal changes, indicating a complex interplay affecting different DOM components. The PCA results also support that in addition to riverine and seawater inputs, in situ primary productivity, degradation processes, and the release/absorption of DOM associated with sediment resuspension also play significant roles in DOM dynamics.

Given the crucial role of DOM in the marine carbon cycle, we recommend further investigations to elucidate DOM compositions and reactivity at varying levels and enhance our understanding of carbon cycling in estuarine and coastal ecosystems.

**Author Contributions:** Conceptualization, Y.X. and Y.W.; methodology, N.P. and Y.W.; software, N.P. and L.L.; validation, N.P., Y.W. and S.P.; formal analysis, N.P.; investigation, N.P. and L.L.; resources, Z.L. and D.D.; data curation, N.P. and Y.W.; writing—original draft preparation, N.P. and Y.W.; writing—review and editing, Y.X.; visualization, N.P. and Y.W.; supervision, Y.W. and Y.X.; project administration, Y.X.; funding acquisition, Y.X. All authors have read and agreed to the published version of the manuscript.

**Funding:** This work was funded by the Shanghai Science and Technology Commission (23230760300) and the National Natural Science Foundation of China (41676058 and 42276033).

**Data Availability Statement:** The data that support this study's findings are available upon request from the corresponding author.

**Acknowledgments:** We thank the captain and crews of the RV Songhang for their support in obtaining samples.

**Conflicts of Interest:** The authors declare no conflicts of interest.

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
