# Peer review of "Tidal Effects on Dissolved Organic Matter Dynamics in a Brackish Water Front Adjacent to Yangtze River Estuary"

_water, doi:10.3390/w17020226_

Round 1
Reviewer 1 Report
Comments and Suggestions for Authors
Dear colleagues,
I am currently reviewing a manusript entitled "Tidal effects on dissolved organic matter dynamics in brackish water front adjacent to Yangtze River Estuary", submitted to Water journal.
The submission is dealing with the assessment the quantity and composition of dissolved organic matter over a 24-hour tidal cycle at the brackish water front near the Yangtze River estuary using various laboratory techniques.
The manuscript is well-structured and written. All the Chapters and Sub-chapters are logical as well as the sequence of material presentation.
I believe that this submission is worth to be published in a desired journal. However, I will leave some comments for consideration by authors:
1. Please try to extend Introuction part to give more detailed information on the relevance of the topic as well as the study area.
2. Please try to avoid citation over-kill (i.e. 14,17,18-20 - line 62; 2-5 - line 35 etc.).
3. Despite, it is up to the author to structure the manunscript, I suggest to try combining the Results and Discussion sections into one Chapter. In my opinion, in this case the manuscript will have higher readability.
4. Please double-check all the Figures for quality to have them readable after the production.
Overall, I wish the authors all the best and quick publication of the manuscript in a desired journal.
Kind regards,
reviewer
Author Response
Reviewer 1
I believe that this submission is worth to be published in a desired journal. However, I will leave some comments for consideration by authors:
Comment 1. Please try to extend introduction part to give more detailed information on the relevance of the topic as well as the study area.
Reply: We thank the reviewer for this comment. We have revised the manuscript in response to the reviewer’s suggestion.
Line 51-62: The Yangtze River is the third longest river in the world and the largest river in China, discharging substantial amounts of freshwater, suspended sediments, and nutrients into the East China Sea [13,14]. Song et al. [15] investigated the variations in DOM and nutrients during spring, moderate, and neap tides at the Yangtze River estuary, revealing significant changes in the concentrations of DOM and monosaccharides in relation to tidal fluctuations. These variations may be attributed to the physical mixing of saline and freshwater masses, the adsorption/desorption of particulate matter, and microbial degradation processes. For example, the extreme flood induced resuspension processes, leading to a substantial input of CDOM molecules[16], while terrestrial organic matter also undergoes changes in molecular composition and structure as it passes through the turbidity zone[17]. These studies suggest that changes in hydrodynamic conditions can lead to alterations in DOM molecules at both spatial and temporal scales in the Changjiang River estuary.
Comment 2. Please try to avoid citation over-kill (i.e. 14,17,18-20 - line 62; 2-5 - line 35 etc.).
Reply: This is a good suggestion. We have revised the manuscript according to to the reviewer’s suggestion by reducing the number of citations to avoid over-citation and retaining only those references that are directly relevant to the study. The specific changes can be found in revised the manuscript.
Line 34-36:In estuarine environments, tidal hydrodynamics exert a profound influence on biogeochemical processes and ecosystems[3-5].
Line 64-67:Despite a relatively comprehensive understanding of the types, structures, and seasonal variations of DOM in the Yangtze River estuary, previous studies have primarily focused on larger spatial scales (tens to hundreds of kilometers) and longer temporal scales (such as seasonal and interannual variations) [18,19], with limited research on rapid changes at smaller scales.
Comment 3. Despite, it is up to the author to structure the manuscript, I suggest to try combining the Results and Discussion sections into one Chapter. In my opinion, in this case the manuscript will have higher readability.
Reply: We appreciate the reviewer’s suggestion. We acknowledge that some journals require the presentation of the Results and Discussion sections separately, whereas others allow to combine Results and Discussion into one session. For our study, the relatively large number of variables, spectral parameters and mass spectrometry data were investigated. Also, some optical and mass spectra parameters like SUVA254, HIX, C1 and AImod are all related to aromatic components in DOM. For example, S275-295 and molecular weight (MW) are parameters that reflect the size of DOM molecules, while AImod and SUVA254 are indicators related to molecular humification. Therefore, it is better to present the result independently, and then discuss by combining different indicators. Therefore, we did not change the structure of our manuscript .
Comment 4. Please double-check all the Figures for quality to have them readable after the production.
Reply: We appreciate the reviewer’s suggestion. We have re-examined the figures and have uploaded the vector graphics to the system.

Reviewer 2 Report
Comments and Suggestions for Authors
Please check the attachement

- Improve the English, grammar, format, scientific words, and typo mistakes.
Author Response
Response to Reviewers
Reviewer 2
Comment 1. Abstract should be concise.
Reply: We have rewritten the abstract and make it more concise. Please see line 9 to 26 in the revised manuscript.
Comment 2. The manuscript needs to be formatted. For example: gram, minutes, hours should be as “g” “min”, “h”; use unique scientific units.
Reply: This is a good suggestion. We have made revisions throughout the manuscript, such as changing 'hours' to 'h' and 'meters' to 'm'.
Comment 3. Authors should avoid too many abbreviations.
Reply: We accepted this suggestion and did not use the abbreviation terms unless they appeared more than three times in the manuscript.
Major Comments
Comment 4 The introduction section needs to be re-organized to provide details in the importance of this study, advantages of this, specify the scientific issues and research gap with the detailed literature data.
Reply: Thank you for this suggestion. We have reorganized the introduction section and added additional references to enhance the logical flow of the article. The revisions are as follows:
The Yangtze River is the third longest river in the world and the largest river in China, discharging substantial amounts of freshwater, suspended sediments, and nutrients into the East China Sea [13,14]. Song et al. [15] investigated the variations in DOM and nutrients during spring, moderate, and neap tides at the Yangtze River estuary, revealing significant changes in the concentrations of DOM and monosaccharides in relation to tidal fluctuations. These variations may be attributed to the physical mixing of saline and freshwater masses, the adsorption/desorption of particulate matter, and microbial degradation processes. For example, the extreme flood induced resuspension processes, leading to a substantial input of CDOM molecules[16], while terrestrial organic matter also undergoes changes in molecular composition and structure as it passes through the turbidity zone[17]. These studies suggest that changes in hydrodynamic conditions can lead to alterations in DOM molecules at both spatial and temporal scales in the Changjiang River estuary.
Despite a relatively comprehensive understanding of the types, structures, and seasonal variations of DOM in the Yangtze River estuary, previous studies have primarily focused on larger spatial scales (tens to hundreds of kilometers) and longer temporal scales (such as seasonal and interannual variations) [18,19], with limited research on rapid changes at smaller scales. This study aims to investigate the brackish water front a site near the Yangtze River estuary, tracking the changes in DOM concentration and composition during tidal cycle, and shedding insight on tidal effect on carbon cycling in the river-dominated margin.
Comment 5 Wha is the impact of this study?
Reply: We thank the reviewer for this comment. We have rewritten the introduction section based on the reviewer’s comments. The impact of our study is that: Although the scientific community has widely acknowledged the critical role of tides in the biogeochemical cycling of estuarine regions, there remains a limited understanding of the rapid variations in organic matter over tidal cycles. This study aims to investigate the brackish water front a site near the Yangtze River estuary, tracking the changes in DOM concentration and composition during tidal cycle, and shedding insight on tidal effect on carbon cycling in the river-dominated margin.
Comment 6 How is this study being different from the reported study?
Reply: Despite a relatively comprehensive understanding of the types, structures, and seasonal variations of DOM in the Yangtze River estuary, previous studies have primarily focused on larger spatial scales (tens to hundreds of kilometers) and longer temporal scales (such as seasonal and interannual variations), with limited research on rapid changes at smaller scales. This study aims to investigate the brackish water front a site near the Yangtze River estuary, tracking the changes in DOM concentration and composition during tidal cycle, and shedding insight on tidal effect on carbon cycling in the river-dominated margin.
Comment 7 What is the real research issue and what are the advantages of this study?
Reply: Our study investigates the variations in DOM in the waters surrounding the Zhoushan Archipelago, influenced by the freshwater discharge from the Yangtze River, over tidal cycles. Utilizing elemental analysis, ultraviolet and fluorescence spectroscopy, and high-resolution mass spectrometry (FT-ICR MS), the results indicate significant differences in the chemical properties of DOM correlated with tidal changes.
Comment 8 What is the novelty of the findings?
Reply: Our work will provide valuable insights into the rapid, tide-driven variations of DOM in the Yangtze River estuary, contributing to a more nuanced understanding of organic carbon cycling in estuarine systems and the broader global carbon cycle.
Comment 9 Authors must mention the study periods, and do they observe any natural weathering? If any, authors should mention it and add the discussion about the influence of these.
Reply: We collected the sampling from May 11 to 12, 2023. During this relatively short time, we did not observe any natural weathering.
Comment 10 Provide the sample river sample information.
Reply: In our study, we did not collect samples from rivers. So we did not provide river information. However, there are a lot of reports on river samples in the Yangtze River estuary, and we cited some of them if necessary in our revised manuscript.
Comment 11 Line no’ 84, how long is the distance from the land?
Reply: We did not measure the distance from the sampling station to the land. Instead, our sampling was based on the location of the turbidity front, which is situated between land and the ocean. This can be reflected in our salinity data.
Comment 12 Have you checked samples analysis at the mid-point of the junction of river and sea water?
Reply: Since our samples were collected at the brackish front near the Yangtze River estuary, we don;t have river samples. In stead, we checked temporal variations in chemical characteristics of DOM during the 24-hour tidal cycle.
Comment 13 Line no’ 89-91, provide the data.
Reply: In situ measurements of physicochemical parameters of surface water, including temperature, pH, and salinity, were conducted using portable instruments. The temperature ranged from 17.2 to 19.6°C, pH from 7.97 to 8.01, and salinity from 25.5 to 28.7 ppt.
Line 94-97: Concurrently, in situ measurements of surface water physicochemical parameters, including temperature (17.2-19.6°C), pH (7.97-8.01), and salinity (25.5-28.7 ppt), were conducted using portable instruments.
Comment 14 Line no’ 95, how much quantity of HCl used (volume & concentration)? Why did you choose this acid rather than other acids?
Reply: In the analysis of marine samples, hydrochloric acid (HCl) is typically employed to eliminate inorganic carbon prior to the measurement of organic carbon. The preference for HCl over other acids is based on its reaction with inorganic carbon, which produces chloride salts that do not interfere with the measurement of other biogeochemical molecules, such as sulfur, phosphorus, and various trace elements. Furthermore, HCl does not oxidize organic matter, in contrast to nitric acid (HNO3) and sulfuric acid (H2SO4). Additionally, unlike phosphoric acid (H3PO4), the byproducts of HCl, specifically chloride salts, are generally soluble in water, whereas phosphate salts may precipitate, potentially leading to inaccuracies in total organic carbon measurements.
Comment 14 What is the ratio of sample and HCl used?
Reply: We have clarified in the methodology section that hydrochloric acid was added until the pH of 2. In solid phase extraction, in order to maximize the sample recovery, it is first necessary to first acidify the filtered specimen with hydrochloric acid to give a pH of 2. It only needs a few drops of concentrated HCl.
Comment 15 Line 99, solid phase extraction by which materials? Provide the complete procedure.
Reply: We thank the reviewer for this comment. Solid Phase Extraction (SPE): DOM is extracted by retaining the target compounds in the sample on an appropriate solid phase and eluting them with a suitable solvent. In SPE, as the DOM sample flows through a pre-packed column, the molecules are retained on the appropriate solid phase and the target compounds are then eluted with a suitable solvent. Solid phase extraction is based on the physical adsorption between the DOM and the solid phase adsorbent. Agilent Bond Elut PPL is a proprietary modified styrene divinylbenzene polymer that is stable at pH extremes and compatible with a wide range of solvents. Agilent Bond Elut PPL 200 mg was used in this experiment.
In solid phase extraction, in order to maximize the sample recovery, it is first necessary to first acidify the filtered specimen with hydrochloric acid to give a pH of 2. This reduces the water solubility of the carboxylic acids and phenolics as well as increases their adsorption capacity, thus increasing the carbon recovery. Afterwards, the salts in the samples were rinsed with acidified pure water (pH=2) to ensure that all the salts in the samples were recovered without affecting the final results. Finally, the recovered salts were then eluted with methanol solution to ensure that the final quality of the samples obtained met the experimental.
Comment 16 Are the conclusions consistent with the evidence and arguments presented and do they address the main question posed? Please also explain why this is/is not the case.
Reply: We thank the reviewer for this comment. We have recheck the conclusion and discussion sections. Some of the evidence we obtained aligns with our conclusions, and we have cited previous literature to support our arguments and viewpoints. As for why we interpret the results in this way, it is based on the consistent conclusions from a large body of established and mature research, which we have referenced in our study.
Comment 17 Results and discussion are too lengthy and not clear.
Reply: Our data include spectra, which consist of both UV and fluorescence, as well as mass spectrometry data. Therefore, when presenting the data, there is a large amount of information. However, all these data are part of the evidence supporting our conclusions, and each piece of data requires analysis and citation of previous literature for validation. This leads to a relatively long text, but every statement supports the conclusions of the paper. As for any unclear sections, we have consolidated and summarized them based on the reviewer’s comments
Comment 18 Tide generation is also dependent on the atmospheric pressure & temperature, wind flow, etc. How did you incorporate these parameters into this study?
Reply: We thank the reviewer for this comment. It is true that tidal processes are influenced by various external conditions. However, our study focuses on the impact of tides on DOM. Therefore, we only observed the changes in DOM during the tidal cycle and discussed the controlling factors of these changes. The external conditions of the tides are beyond the scope of the current study. Nevertheless, we acknowledge these factors like atmospheric pressure should be considered in future studies.
Comment 19 Sec 2.1, how did you derive this conclusion? Elaborate this section.
Reply: We thank the reviewer for this comment. We are not entirely sure which conclusion the reviewer is referring to. However, regardless of which aspect the reviewer means, we have carefully re-examined the conclusion section of our paper and are confident that our data support these conclusions. Moreover, these findings are consistent with the conclusions of previous studies.
Comment 20 Sec 2.2, Line 201, “likely due to strong hydrodynamic forces that increased disturbances between the water column and surface sediments,”, what are they?
Reply: We are sorry for this confusion. For a325, high-turbidity terrestrial water bodies and bottom-layer suspended waters can lead to an increase in values. These findings are supported by previous studies, such as Reference [16,17]. Their research indicates that the resuspension process in the water can enhance the input of some terrestrially-derived substances. Therefore, in our paper, we interpret this as possibly due to the strong hydrodynamics increasing the disturbance between the water column and surface sediments, which results in a significant increase in a325 values.
Comment 21 Authors incorporate the basic soil test in the study area, atmospheric pressure & temperature, wind flow etc?
Reply: We thank the reviewer for this comment. Our study is based on seawater and investigates the changes in DOM under tidal influences. Changes in external soil properties or weather conditions may indeed affect DOM variations, but these factors are not the focus of this paper. We will consider the reviewer’s suggestion and incorporate this aspect into our future research.
Comment 22 The reviewer is stopped at this point due to their way of construction of the manuscript and presentation of the data discussion. It is hard to follow-up on their manuscript. Authors must improve their presentation, writing, and English.
Resply: We respectfully disagree with this comment. We engaged several senior experts to refine our manuscript. Additionally, the corresponding author possesses over 20 years of academic experience and has published more than 100 articles in reputable journals such as Nature Communications, PNAS, Environmental Science & Technology, and Journal of Geophysical Research, among others. Therefore, we are confident in our proficiency in English writing.
Reviewer 3 Report
Comments and Suggestions for Authors
The authors of the manuscript "Tidal effects on dissolved organic matter dynamics in brackish water front adjacent to Yangtze River Estuary" have studied, in their manuscript, how dissolver organic carbon matter concentration could be influenced by the tydal cycle near the Yangtze River Estuary.
Study area and sampling and pretreatment were widely characterized. Analytical methods were detailed and well described and were fully consistent with the discussion of experimental data showed in the manuscript.
The relationship between salinity and tidal variation was accurately investigated and Figure 2 clearly describes it. The variation in fluorescence characteristics was also deeply studied and clearly depicted in Figure 3. The chemical composition (% CHO, % CHONS and others) was also subjected to an intensive analysis to find out correlations with samples near high and low tide. Then, the characteristics of dissolved organic matter and total nitrogen parameters were discusses and explained along with the tidal cycle.(Figure 6).
References and conclusions are consistent with the findings and the data elaboration showed in the manuscript.
Under these considerations, my overall recommendation is to "accept in present form".
Author Response
Reviewer 3
The authors of the manuscript "Tidal effects on dissolved organic matter dynamics in brackish water front adjacent to Yangtze River Estuary" have studied, in their manuscript, how dissolver organic carbon matter concentration could be influenced by the tydal cycle near the Yangtze River Estuary.
Study area and sampling and pretreatment were widely characterized. Analytical methods were detailed and well described and were fully consistent with the discussion of experimental data showed in the manuscript.
The relationship between salinity and tidal variation was accurately investigated and Figure 2 clearly describes it. The variation in fluorescence characteristics was also deeply studied and clearly depicted in Figure 3. The chemical composition (% CHO, % CHONS and others) was also subjected to an intensive analysis to find out correlations with samples near high and low tide. Then, the characteristics of dissolved organic matter and total nitrogen parameters were discusses and explained along with the tidal cycle.(Figure 6).
References and conclusions are consistent with the findings and the data elaboration showed in the manuscript.
Under these considerations, my overall recommendation is to "accept in present form".
Reply: We sincerely thank Reviewer 3 for the kind words regarding our paper. However, we are aware that there are still some minor errors in the article. Following the suggestions of Reviewer 1 and Reviewer 2, we have made improvements to the paper based on their feedback.
Round 2
Reviewer 2 Report
Comments and Suggestions for Authors
The manuscript is NOT satisfactorily improvised.
Comments on the Quality of English LanguageThe response to this query "The reviewer is stopped at this point due to their way of construction of the manuscript and presentation of the data discussion. It is hard to follow-up on their manuscript. Authors must improve their presentation, writing, and English.", especially about the English, the authors are poorly understand the question, they haven't checked their manuscript and improvised.